# Towards Novel Metamaterial Discovery via Latent Space Regulation and Exploration

## Abstract

Metamaterials are artificially engineered structures whose unique mechanical and physical properties arise from geometry rather than composition, enabling applications in wave control, energy absorption, and soft robotics. To capture this structural programmability in a unified form, voxel representation provides a natural choice: it can express diverse classes of metamaterials including truss, shell, and porous metamaterials within a single cubic discretization. However, existing voxel-based generative models face severe limitations. The vast design space, combined with sparse and costly datasets, leads to a generalization dilemma: models tend either to memorize known designs, sacrificing novelty, or to produce invalid, low-quality structures. To address this, we propose **VoxPlorer**, a generative framework that couples voxel representation with latent space regulation and guided exploration. VoxPlorer introduces a repel-and-sink (RAS) mechanism to smooth and densify the latent distribution of valid structures, and a short-range repulsion (SRR) guidance during diffusion to promote exploration beyond memorized regions while preserving validity. We further contribute a systematic benchmark for voxel-based metamaterials and develop an evaluation module that jointly assess quality, novelty, and diversity. Extensive experiments show that VoxPlorer outperforms state-of-the-art baselines, achieving +8.9% in quality, +46.4% in novelty, and +128.6% in diversity on average across two datasets, establishing a principled pathway toward systematic discovery of next-generation metamaterials.

## 1 Introduction

Metamaterials are artificially engineered structures whose unusual behaviors arise from carefully designed geometries rather than intrinsic chemical composition. This structural programmability enables properties rarely observed in natural materials, such as negative Poisson's ratio, ultrahigh stiffness-to-weight ratio, and extreme energy absorption (Zhang et al., 2016; Mizzi & Spaggiari, 2020). These capabilities have driven breakthroughs across domains including biomedical scaffolds, vibration isolation, acoustic cloaking, soft robotics, and thermal management (Bertoldi et al., 2017; Liu & Zhang, 2011). The ability to tailor functionality at the micro- and meso-scale positions metamaterials as a critical frontier for next-generation engineering systems.

Given their extraordinary potential, metamaterials have become a rising focus in material science over the past two decades (Kadic et al., 2019). Early design efforts relied heavily on human expertise and manual construction, but the emergence of machine learning has enabled data-driven approaches to accelerate discovery. Existing methods largely fall into two categories: modeling metamaterials as 3D graphs (Zhan et al., 2025; Bastek et al., 2022; Maurizi et al., 2025), or designing 2D patterns that are extruded uniformly along a third axis (Kollmann et al., 2020; Tian et al., 2022; Wilt et al., 2020). Graph representations provide an abstract and interpretable view of metamaterials, yet they lack the ability to express fine-grained geometric details, as edges are usually instantiated as simple primitives like cylinders or cuboids. In contrast, 2D pattern-based designs construct a repeating planar motif and then extrude it uniformly along the third axis to form a 3D structure. Such designs can achieve superior performance in the two in-plane directions defined by the patterned motif, but along the extruded axis the properties remain largely unchanged from the base material, offering little improvement. Recently, voxel representation (i.e., discretizing a cubic space into small cells marked as void or solid) has become an arising research direction for metamaterial. There are only a few attempts (Zheng et al., 2023b; 2025; Yang et al., 2024) following this direction by naively adapting 3D generation models from computer vision domain to the metamaterial domain, without specifically catering the need and nature of metamterial. Therefore, this research direction is still

largely under-explored. Unlike other modalities that are tailored to specific classes of metamaterials (e.g., graphs for trusses, images for 2D designs), voxel representation serves as a unified format capable of expressing all kinds of metamaterials, including truss-based, shell-based, porous, 2D, kirigami metamaterials, etc. This makes it a compelling modality for comprehensive design and evaluation.

Despite its promise, voxel representation also introduces a major challenge: the generalization dilemma. For example, at a resolution of $64$ per axis, there are $2^{64^3}$ possible configurations. However, against the high number of configurations, voxel datasets are costly to build and store and therefore are limited in size (around 10,000). As a result, valid designs occupy only a tiny fraction of the voxel space, preventing machine learning models from smoothly approximating the underlying distribution. Consequently, generative models tend either to memorize training samples and lose novelty, or to produce invalid outcomes such as pure voids or solids. Current voxel-based approaches remain limited: some adapt diffusion models to the voxel domain (Zheng et al., 2025; Yang et al., 2024), while others employ generative adversarial networks (Zheng et al., 2023b). These methods attempt to approximate valid distributions directly, without explicitly addressing sparsity, and thus remain vulnerable to the generalization dilemma.

Formally, we identify two key challenges for voxel-represented metamaterial design. **(C1. Generalization Dilemma):** the vast design space versus limited training data forces generative models to either reproduce seen samples with high fidelity or produce unseen ones of poor quality (as observed in our experiments). **(C2. Lack of Benchmark):** to the best of our knowledge, only Yang et al. (2024) provides a public dataset of shell-type metamaterials, which is large enough to train deep generative models. However, there are various metamaterials of other types, like truss-based metamaterials. Besides, there is also a lack of comprehensive evaluation system. Therefore, a systematic benchmark is needed to support voxel-represented metamaterial design.

To address **C1**, we propose **VOXPLORER**, a generative framework that combines latent regulation and exploration. VOXPLORER encodes voxel structures into a low-dimensional latent space via an autoencoder, then applies a novel repel-and-sink (RAS) mechanism to smooth and densify the distribution of valid samples, mitigating sparsity and enhancing validity. To further promote generalization, we introduce short-range repulsion (SRR) guidance into the diffusion process, which discourages generation near memorized samples and drives exploration into less-populated regions of the design space. To address **C2**, we construct so far as we know the first publicly available voxel dataset of considerably large size for truss-based metamaterials and propose five metrics to jointly evaluate quality, novelty, and diversity.

Through extensive experiments on our dataset and the dataset from Yang et al. (2024), we show that VOXPLORER significantly outperforms state-of-the-art voxel-based baselines, improving quality by $8.9\%$, novelty by $46.4\%$, and diversity by $128.6\%$ on average across both datasets. Additional analyses and visualizations of the latent space and generated structures further verify the effectiveness of RAS for latent regulation and SRR for exploratory generation.

## 2 PRELIMINARY

This section introduces the concept of metamaterials, their voxel representation, relevant generative models, and the overall problem definition.

### 2.1 VOXEL-REPRESENTATION FOR METAMATERIALS

Metamaterials are artificial micro-structures composed of substrate material (e.g., plastics, metals, ceramics). Their geometry can be naturally described by unit cell $\mathbf{U}$ and lattice $\boldsymbol{l} = (l_x, l_y, l_z) \in \mathbb{R}^3$, where $\mathbf{U}$ defines the distribution of substrate material within a cube or cuboid unit, and $\boldsymbol{l}$ specifies repetition intervals along the $x$, $y$, and $z$ axes (Figure 1). A design can therefore be denoted as $\mathcal{M} = (\mathbf{U}, \boldsymbol{l})$. In this work, $\mathbf{U}$ is expressed in voxel form

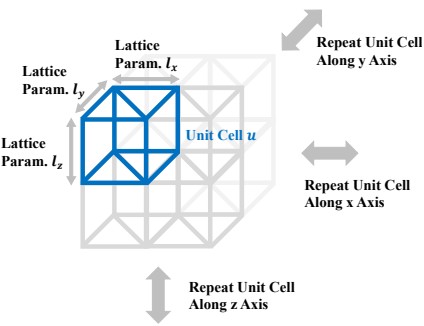

Figure 1: Unit cell and lattice of metamaterials.

as a binary tensor $\mathbf{U} \in \mathbb{B}^{d^3}$ where $\mathbb{B} = \{0, 1\}$ and $d$ is the voxel resolution. We denote the dataset of unit cell samples $\mathbf{U}$ as $\mathcal{U}$.

## 2.2 RELATED MODELS FOR VOXEL GENERATION

**Autoencoders (AEs).** An AE maps voxel data to a latent space via an encoder $\mathcal{E}$ and reconstructs it with a decoder $\mathcal{D}$. Training minimizes reconstruction loss:

$$L_{\text{recon}} = \frac{1}{N} \sum_{i=1}^{N} ||\mathcal{D} \circ \mathcal{E}(\mathbf{U}_i) - \mathbf{U}_i||, \tag{1}$$

where $\mathbf{U}_i$ is the $i$th voxel-represented sample, $N$ the dataset size, and $\cdot \circ \cdot$ is the composition of two functions. The latent variable is $\boldsymbol{x}_i = \mathcal{E}(\mathbf{U}_i)$. To enable generation, the latent distribution $\boldsymbol{x}$ must be specified or approximated. For instance, variational AEs (VAEs, Kingma & Welling (2013)) regularize $\boldsymbol{x}$ to a Gaussian, sampling $\boldsymbol{x} \sim \mathcal{N}(0, 1)$ for decoding.

**Diffusion Models.** Diffusion connects arbitrary data distributions with Gaussians via reverse denoising. Following DDPM (Ho et al., 2020), each denoising step can be expressed as:

$$\boldsymbol{x}_{t-1} = \frac{1}{\sqrt{1 - \beta_t}} \left( \boldsymbol{x}_t - \frac{\beta_t}{\sqrt{1 - \alpha_t^2}} \phi_{\text{diff}}(\boldsymbol{x}_t, t) \right) + \rho_t \boldsymbol{\epsilon}, \tag{2}$$

where $\phi_{\text{diff}}$ is the diffusion model, $\boldsymbol{\epsilon}$ Gaussian noise, and $\alpha_t, \beta_t, \rho_t$ hyperparameters. Diffusion can also model latent spaces, referred as latent diffusion models (Rombach et al., 2022).

## 2.3 PROBLEM DEFINITION

Given challenge of the generalization dilemma, the problem we are tackling is to generate samples preserving both quality and novelty to the largest extent. Besides, since it is not ideal for the generation results to converge to only a few similar samples, we also need to consider the diversity of generated samples. This can be expressed as a multi-objective problem where the quality, novelty and diversity are regarded as design objectives. Quality evaluates the degree to which a sample is valid, usually appearing as the periodicity, symmetry and connectivity of the structure. Novelty evaluates how much a sample is different from known samples, so it is defined with respect to a certain dataset. Diversity evaluates how different the samples are from each other, measured as the fraction of design space covered by the generated samples. In summary, the problem is as follows:

**Problem Definition.** Let $f$ denote a generative model defined as $\mathbf{U} = f(\boldsymbol{x})$. The objective is to identify an $f$ that maximizes performance across all the three aspects: quality, novelty, and diversity.

## 3 BENCHMARK DEVELOPMENT

To the best of our knowledge, there is only one publicly available voxel-represented metamaterial dataset from Yang et al. (2024). Besides, the evaluation for generated metamaterial designs are mostly conducted by visualization and human assessment, so a comprehensive evaluation framework is still lacking. To enable a systematical study of metamaterial design in voxel-representation, we propose the first systematical benchmark to provide data support and result evaluation for voxel-represented metamaterial design.

Figure 2: Benchmark Development.

## 3.1 DATASET DEVELOPMENT

We propose a unified voxel-based representation for metamaterial datasets. To the best of our knowledge, the only existing voxel dataset is **MetaShell** (Yang et al., 2024), which contains shell-type metamaterials where unit cells are defined by curved surfaces. While valuable, this dataset covers

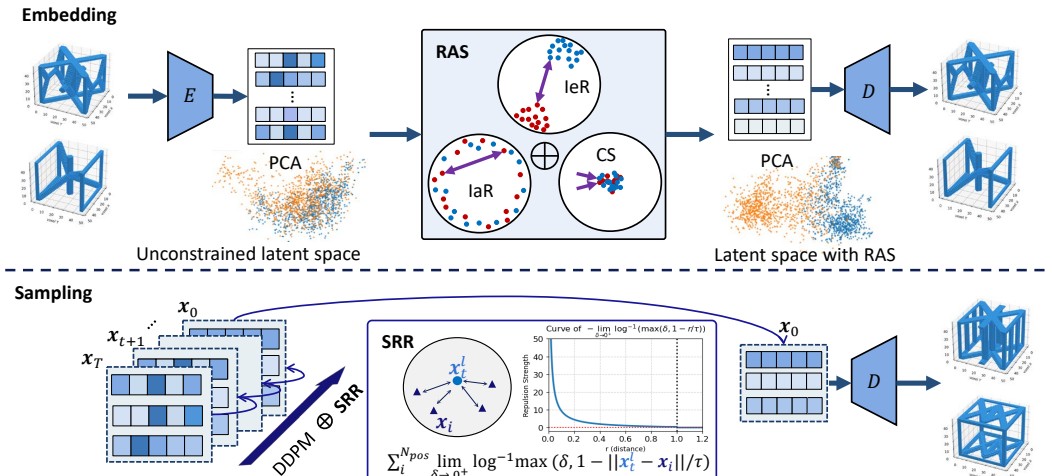

Figure 3: An overview of the proposed framework VoxPlorer. It encodes voxel structures into latent space, applies RAS for latent regulation, and employs SRR-guided diffusion to explore and decode novel yet valid metamaterials.

only one class of designs. Truss-based metamaterials represent another critical category for mechanical applications (Mizzi & Spaggiari, 2020; Song et al., 2025), yet existing truss datasets rely on graph representations (Lumpe & Stankovic, 2021; Bastek et al., 2022), which are natural for describing connectivity but lack fine-grained geometric detail. Reformatting truss structures into voxel space not only unifies them with shell-type metamaterials under a common representation, but also enables richer structural details. To close this gap, we construct the first truss-based voxel dataset, which we call **MetaTruss**. MetaTruss is derived from (Lumpe & Stankovic, 2021), where original samples are provided in 3D graph form. Each unit cell is discretized into a $48^3$ voxel grid: voxels lying within a truss radius of any graph edge are marked as solid, while all others remain void. Following this procedure, we process the first 10,000 samples from (Lumpe & Stankovic, 2021) and apply rotational augmentation to expand the dataset to 60,000 samples. Together with MetaShell reformatted into voxel space, our benchmark establishes a unified data module that remains compatible with future metamaterial datasets. More details are in Appendix C.

## 3.2 Evaluation Mechanism

To systematically evaluate the generated metamaterial structures, we propose five metrics from three aspects. **Quality Scores:** we propose symmetry score $S_{\mathrm{sym}}$ to evaluate central symmetry degree of a structure; periodicity score $S_{\mathrm{per}}$ to evaluate how similar each facet of the cube frame is to its parallel counterpart; connectivity score $S_{\mathrm{con}}$ to evaluate how well the structure is connected, i.e., the fraction of the largest connected bulk in $U_{\mathrm{solid}}$. **Novelty Score**: we propose $S_{\mathrm{nov}}$ to evaluate the IoU distance between a sample and its nearest neighbor in the training dataset. **Diversity Score:** we propose $S_{\mathrm{div}}$ to evaluate how many different samples in the training dataset function as a nearest neighbor of a generated sample, and divide this number by the number of generated samples. More details regarding the dataset and metric development can be found in Appendix C.

## 4 Methodology

This section introduces our framework, VoxPlorer. We begin with a high-level overview, then describe the autoencoder that maps voxel structures into a low-dimensional latent space. We next present the RAS mechanism, which regularizes this space to separate valid and invalid regions, followed by a latent diffusion process with SRR that promotes exploration for novel designs. An optional refinement module further improves voxel-level quality. Each subsection details the purpose and technical design of these components.

### 4.1 Framework Overview.

The main challenge in voxel-based metamaterial generation is the **generalization dilemma (C1)**: models trained on a small fraction of the design space because of costly and limited data either overfit to training samples, losing novelty, or produce invalid structures when exploring. VoxPlorer tackles this in two steps. First, an autoencoder maps voxels into a compact latent space regulated by the

RAS mechanism, which separates valid from invalid regions, prevents mode collapse, and densifies the feasible manifold. This alleviates the influence of small perturbations on decoding the latents, improving robustness and generalization. Second, a latent diffusion model with SRR guidance discourages samples too close to training data, pushing exploration toward less-populated yet feasible regions. Together, RAS provides stability and SRR drives exploration, directly addressing C1 and enabling VoxPlorer to balance quality, novelty, and diversity.

### 4.2 Autoencoding with RAS Latent Regulation

To mitigate the high dimensionality of voxel representation, we use an AE to compress voxel data into a low-dimensional latent space. However, simply regularizing the latent distribution (as in VAEs) often reduces generation quality, since metamaterials must satisfy strict constraints like periodicity, where even slight structural deviations (e.g., isolated "floating" clusters) are invalid. Prior AE-based methods lack tailored latent regulation, resulting in a latent space where valid and invalid regions are entangled. Such entanglement may lead to failed designs. To address this issue, we propose the RAS mechanism, which disentangles valid and invalid regions in the latent space through three

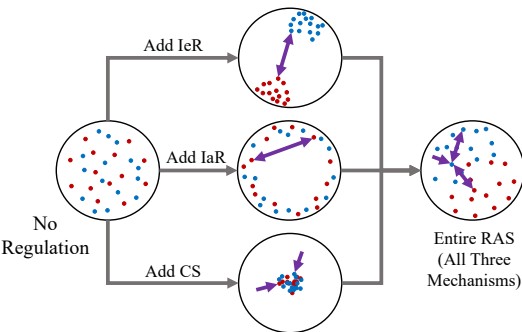

Figure 4: Illustration of the effect of RAS and its components.

component mechanisms: **inter-class repulsion (IeR)**, **intra-class repulsion (IaR)**, and **central sink (CS)**. We first synthesize negative voxel samples from ground-truth structures to train the separation. Let $\mathcal{U}_{\text{pos}}$, $\mathcal{U}_{\text{neg}}$, and $\mathcal{U} = \mathcal{U}_{\text{pos}} \cup \mathcal{U}_{\text{neg}}$ denote the positive, negative, and full datasets. Encoding $\mathcal{U}$ with $\mathcal{E}$ yields latent datasets $\mathcal{X}$, with $\mathcal{X}_{\text{pos}}$ and $\mathcal{X}_{\text{neg}}$ denoting the positive and negative subsets.

**Inter-Class Repulsion.**[1] IeR mechanism aims to simplify the decision boundary between $\mathcal{X}_{\text{pos}}$ and $\mathcal{X}_{\text{neg}}$ by adding inverse-square repulsion similar to Coulomb repulsion (Anisimov et al., 2009):

$$F_{\text{inter}}(\mathcal{X}_{\text{pos}}, \mathcal{X}_{\text{neg}}) = \sum_{i=1}^{|\mathcal{X}_{\text{pos}}|} \sum_{j=1}^{|\mathcal{X}_{\text{neg}}|} \frac{\boldsymbol{x}_{\text{pos},i} - \boldsymbol{x}_{\text{neg},j}}{||\boldsymbol{x}_{\text{pos},i} - \boldsymbol{x}_{\text{neg},j}||^3}, \tag{3}$$

where $\boldsymbol{x}_{\text{pos},i}$ and $\boldsymbol{x}_{\text{neg},j}$ are the $i$th positive latent sample and $j$th negative latent sample, respectively. The simulated distribution of adding IeR alone can be found in Figure 4. In order to use IeR to optimize the latent distribution, we can minimize the integral of $F_{\text{inter}}$, i.e., the Coulomb potential:

$$P_{\text{inter}}(\mathcal{X}_{\text{pos}}, \mathcal{X}_{\text{neg}}) = \sum_{i=1}^{|\mathcal{X}_{\text{pos}}|} \sum_{j=1}^{|\mathcal{X}_{\text{neg}}|} ||\boldsymbol{x}_{\text{pos},i} - \boldsymbol{x}_{\text{neg},j}||^{-1}. \tag{4}$$

**Intra-Class Repulsion.** As illustrated in Figure 4, using IeR alone can simply the decision boundary, but it drives the two classes into two distant clusters. In this case, only a small portion of the latent space is covered, so the decoder may not generalize to the majority of the latent space. Therefore, when the latent generator touches the region out of the two clusters, the decoder will not be able to decode the latent well. To solve this problem, unlike contrastive learning which tries to bring positive latents to each other even more closely (Saunshi et al., 2019), we propose IaR mechanism to avoid the converging tendency of each cluster. Similar to IeR, IaR and its potential are:

$$F_{\text{intra}}(\mathcal{X}_{\text{pos/neg}}) = \sum_{i,j=1;i\neq j}^{|\mathcal{X}_{\text{pos/neg}}|} \frac{\boldsymbol{x}_{\text{pos/neg},i} - \boldsymbol{x}_{\text{pos/neg},j}}{||\boldsymbol{x}_{\text{pos/neg},i}s - \boldsymbol{x}_{\text{pos/neg},j}||^3}. \tag{5}$$

$$P_{\text{inter}}(\mathcal{X}_{\text{pos/neg}}) = \sum_{i,j=1;i<j}^{|\mathcal{X}_{\text{pos/neg}}|} ||\boldsymbol{x}_{\text{pos/neg},i} - \boldsymbol{x}_{\text{pos/neg},j}||^{-1} \tag{6}$$

---

[1]In this paper $|\cdot|$ means the sum of all elements if a tensor is inside (e.g., $\mathbf{U}$), or the cardinality of a set when a set is inside (e.g., $\mathcal{U}$).

**Central Sink (CS).** IeR and IaR will simplify the latent decision boundary and avoid intra-class convergence, but they will also force the latents to be too far from each other, which will sparsify the latent distribution. In order to alleviate this problem, we propose CS mechanism to attract all the latents to the origin. Let $\boldsymbol{x}_i$ be the $i$th latent variable, the force and potential of CS are:

$$F_{\text{sink}}(\mathcal{X}) = \sum_{i=1}^{|\mathcal{X}|} \boldsymbol{x}_i, \text{ and } P_{\text{sink}}(\mathcal{X}) = \sum_{i=1}^{|\mathcal{X}|} ||\boldsymbol{x}_i||^2. \tag{7}$$

The effect of adding CS alone are also shown in Figure 4. When combining all three mechanisms together, we use different coefficients to tune strength of them. Note that the IaR should be significantly weaker than the IeR, or a latent sample will experience strong repulsion from both its "peers" and "opponents", which means the decision boundary will not be simplified. The strength of CS, however, had a less strict constraint and has a large valid range. A stronger CS will make the absolute value of the latents smaller, vice versa. The RAS loss can therefore be defined as:

$$L_{\text{RAS}} = \lambda_{\text{inter}} P_{\text{inter}}(\mathcal{X}_{\text{pos}}, \mathcal{X}_{\text{neg}}) + \lambda_{\text{intra}}(P_{\text{intra}}(\mathcal{X}_{\text{pos}}) + P_{\text{intra}}(\mathcal{X}_{\text{neg}})) + \lambda_{\text{sink}} P_{\text{sink}}(\mathcal{X}), \tag{8}$$

where $\lambda_{\text{inter}}$, $\lambda_{\text{intra}}$ and $\lambda_{\text{sink}}$ are hyperparameters. Then we can obtain the total loss function for training the autoencoder:

$$L_{\text{auto}} = \lambda_{\text{recon}} L_{\text{recon}} + \lambda_{\text{RAS}} L_{\text{RAS}}, \tag{9}$$

where $L_{\text{recon}}$ is defined in Equation 1, $\lambda_{\text{recon}}$ and $\lambda_{\text{RAS}}$ are two hyperparameters.

Combining the three mechanisms into the entire RAS mechanism will regulate the latent space derived by AE so that the valid and invalid region are disentangled and pushed away from each other, making the valid samples in a dense distribution while still maintaining some distance between valid samples to avoid latent space clapse with all the latents converging to a small region.

### 4.3 LATENT DIFFUSION WITH SRR GUIDANCE

The RAS mechanism regulates the latent distribution so that the latent decision boundary is simplified, which means the decoder is more robust to variations of latents around the positive latent sample cluster. In this case, we can use diffusion models to approximate the latent distribution and connect it with a Gaussian distribution for sampling's sake. However, the vanilla diffusion paradigm like DDPM, without further guidance, can still fall into the generalization challenge because the seen samples are usually in the region where the generation probability is high and hence will tend to be reproduced by the diffusion model.

To address this issue, we introduce a repulsion force between the sample being generated and the latents of known samples, guiding the diffusion process toward unexplored regions of the latent space where novel designs may emerge. Crucially, this repulsion must be short-ranged: if it extends too far, the generated sample could be pushed entirely out of the valid region, resulting in invalid outputs. Building on this idea, we propose the SRR mechanism (Figure 3), which augments the DDPM model with an additional SRR guidance term. Based on Equation 2, the denoising step of an SRR-guided DDPM can be expressed as:

$$\boldsymbol{x}_{t-1} = \frac{1}{\sqrt{1-\beta_t}} \left( \boldsymbol{x}_t - \frac{\beta_t}{\sqrt{1-\alpha_t^2}} \phi_{\text{diff}}(\boldsymbol{x}_t, t) \right) + \rho_t \boldsymbol{\epsilon}$$

$$+ \lambda_{\text{SRR}} \sum_{i=1}^{N_{\text{pos}}} \frac{\boldsymbol{x}_{\text{pos},i} - \boldsymbol{x}_t}{||\boldsymbol{x}_{\text{pos},i} - \boldsymbol{x}_t||} \lim_{\delta \to 0^+} \log^{-1} \max(\delta, 1 - ||\boldsymbol{x}_t - \boldsymbol{x}_{\text{pos},i}||/\tau), \tag{10}$$

where $\lambda_{\text{SRR}}$ and $\tau$ are two hyperparameters controlling the SRR guidance strength. The SRR guidance (last term in Equation 10) introduces an additional force that pushes the latent variable $\boldsymbol{x}_t$ away from the known latents. The logarithmic formulation ensures that this repulsion decays rapidly with distance, aligning with the intuition that only nearby samples should have influence. In practice, computing distances to all known latents is prohibitively expensive, so we cluster the known latents before training the diffusion model. During denoising, clusters outside the neighborhood of the current latent variable are ignored by the SRR mechanism, reducing computational cost. Importantly, SRR guidance is applied only at inference for latent space exploration, while the training scheme follows the standard DDPM paradigm.

To sum up, we introduce the RAS mechanism into the functioning flow of AE to obtained a latent space where the valid latents are densely and smoothly distributed and away from invalid latents,

Table 1: Performance Evaluation of Different Approaches

| Approaches | Quality Scores | | | | Novelty Score | Diversity Score |
|---|---|---|---|---|---|---|
| | $S_{\text{sym}}\uparrow$ | $S_{\text{per}}\uparrow$ | $S_{\text{con}}\uparrow$ | Mean $\uparrow$ | $S_{\text{nov}}\uparrow$ | $S_{\text{div}}\uparrow$ |
| **MetaTruss** (ours) | | | | | | |
| DiT-3D (Mo et al. (2023)) | 0.358 | 0.248 | 0.500 | 0.369 | 0.003 | 0.010 |
| Y. Yang et al. (Yang et al. (2024)) | **0.800** | **0.585** | 0.494 | 0.626 | 0.163 | 0.158 |
| XCube (Ren et al. (2024)) | 0.506 | 0.525 | 0.522 | 0.518 | 0.000 | 0.004 |
| Trellis (Xiang et al. (2025)) | 0.081 | 0.063 | 0.133 | 0.092 | 0.000 | 0.001 |
| 3D-CDM (Zheng et al. (2025)) | 0.470 | 0.270 | **0.999** | 0.580 | 0.000 | 0.001 |
| VOXPLORER (ours) | 0.718 | 0.487 | 0.969 | **0.725** | **0.296** | **0.420** |
| **MetaShell** (Yang et al., 2024) | | | | | | |
| DiT-3D (Mo et al. (2023)) | 0.465 | 0.259 | 0.704 | 0.476 | 0.023 | 0.013 |
| Y. Yang et al. (Yang et al. (2024)) | 0.922 | 0.791 | 0.991 | 0.901 | 0.342 | 0.409 |
| XCube (Ren et al. (2024)) | 0.522 | 0.523 | 0.526 | 0.524 | 0.000 | 0.005 |
| Trellis (Xiang et al. (2025)) | 0.795 | 0.576 | **0.999** | 0.790 | 0.103 | 0.032 |
| 3D-CDM (Zheng et al. (2025)) | 0.668 | 0.529 | **0.999** | 0.732 | **0.390** | 0.113 |
| VOXPLORER (ours) | **0.923** | **0.856** | 0.978 | **0.919** | 0.380 | **0.783** |

which facilitates the generation process. In such a latent space, we use latent diffusion with SRR guidance to drive the latent variable away from being very close to known samples, so that the chance of generating novel samples by latent space exploartion is increased.

## 5 EXPERIMENTS

In this section, we evaluate VOXPLORER for its ability to generate high-quality and novel metamaterials. We compare against state-of-the-art baselines using quality, novelty, and diversity metrics, and present ablation studies on the RAS regulation and SRR guidance. Finally, we analyze model capacity sensitivity and visualize generated samples to show the achieved quality-novelty balance.

### 5.1 OVERALL COMPARISON

We evaluate VOXPLORER against five voxel-based generative baselines: DiT-3D (Mo et al., 2023), Yang et al. (Yang et al., 2024), XCube (Ren et al., 2024), Trellis (Xiang et al., 2025), and 3D-CDM (Zheng et al., 2025). The experiments are conducted on our proposed benchmark, which includes the MetaTruss and MetaShell datasets, and the task is unconditional voxel-based metamaterial generation. Performance is assessed using five complementary metrics covering three dimensions: structural quality ($S_{\text{sym}}$, $S_{\text{per}}$, $S_{\text{con}}$), novelty ($S_{\text{nov}}$), and diversity ($S_{\text{div}}$). All models are trained and evaluated on a single NVIDIA A100 GPU (except XCube whose large model size requires two A100 GPUs). To compare the generation capability of VOXPLORER with other baseline models, we train each model on each of the two datasets and compute the five metrics for the generation results. Specific results are shown in Table 1.

Across both MetaTruss and MetaShell datasets, VOXPLORER achieves the best balance of quality, novelty, and diversity. On MetaTruss, it delivers competitive quality while substantially outperforming baselines in novelty and diversity, avoiding the memorization observed in prior methods. On MetaShell, it matches or exceeds state-of-the-art quality and more than doubles diversity, showing that RAS ensures validity and SRR promotes exploration. Together, these results confirm that VOXPLORER resolves the quality–novelty trade-off better than existing voxel-based approaches.

### 5.2 ABLATION STUDY

The experiments in this section is done on MetaTruss, with more results on MetaShell available in Appendix D.

**Latent Space Regulations Comparison.** To verify the effect of our proposed RAS mechanism, we train the autoencoder and visualize the latent space (after compressed by principal component analysis (PCA)), under three different cases: (1) autoencoder without latent regulation; (2) with contrastive regulation (Saunshi et al., 2019); (3) with RAS regulation. The latent distribution in the three cases are shown in Table 2. From the first column of Table 2, we can see that without any regulations, the autoencoder will not distinguish between the positive and negative latents, so that the distributions of these two classes will be mingled together, hence harming the generation quality.

Table 2: Latent distribution visualization and generated samples with RAS regulation, or contrastive regulation, or without regulation. Reg. denotes regulation, and Contra. denotes contrastive.

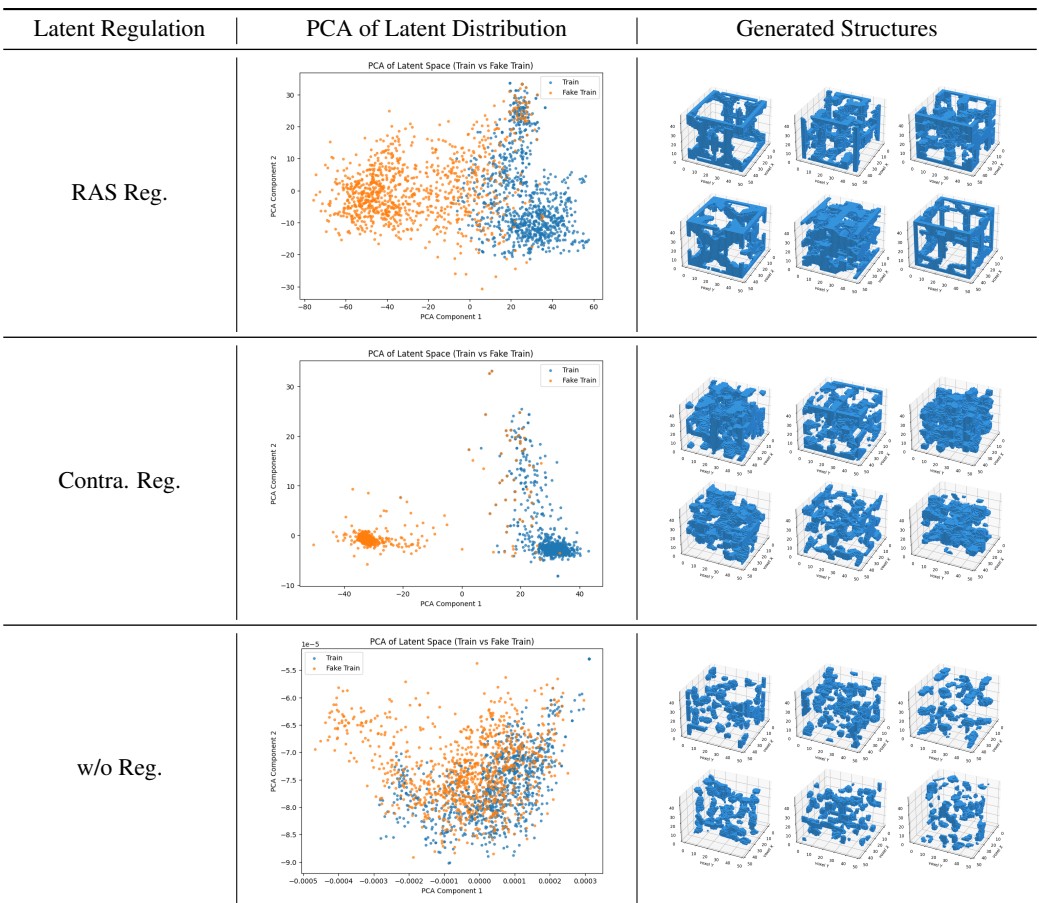

| Latent Regulation | PCA of Latent Distribution | Generated Structures |
|---|---|---|
| RAS Reg. | | |
| Contra. Reg. | | |
| w/o Reg. | | |

With contrastive regulation, latents within the same class are pulled tightly together, while samples from different classes are pushed into two distant regions. Therefore, the two classes converge towards two small regions of the entire latent space. With our proposed RAS, the positive and negative latents are well separated, while the bulk of the latent space is smoothly distributed to increase the robustness of decoding the latents. In case (1) and (2), the decoded voxel structures are of poor quality. In contrast, RAS produces diverse and relatively well-formed geometries, showing that robust latent separation improves quality and diversity of generation. Besides visualization, results in the second row of Table 3 verify the effectiveness quantitatively.

Table 3: Ablation on RAS regulation and SRR diffusion.

| Approaches | Quality Scores | | | | Novelty Score | Diversity Score |
|---|---|---|---|---|---|---|
| | $S_{\text{sym}} \uparrow$ | $S_{\text{per}} \uparrow$ | $S_{\text{con}} \uparrow$ | Mean $\uparrow$ | $S_{\text{nov}} \uparrow$ | $S_{\text{div}} \uparrow$ |
| Case 1 (RAS + vanilla DDPM) | 0.753 | 0.632 | 0.885 | **0.757** | 0.208 | 0.336 |
| Case 2 (w/o reg + SRR Diff.) | **0.873** | **0.801** | 0.295 | 0.656 | 0.014 | 0.011 |
| Case 3 (full framework) | 0.718 | 0.487 | **0.969** | 0.725 | **0.296** | **0.420** |

**SRR diffusion v.s. vanilla DDPM.** Our high level aim is to explore the design space to generate high quality and novel samples. To verify the exploration performance of SRR diffusion, we compare two cases: (1) RAS + vanilla DDPM model; (2) RAS + SRR diffusion (our full framework). We train the model under these two settings and the results are shown in Table 3. From Table 3 we can see that SRR diffusion provides far higher novelty score and diversity score, while maintaining the three quality scores close to vanilla DDPM. The deterioration of generation quality is reasonable because vanilla DDPM tends to generate what the model has seen during training, therefore the

quality can be higher. The results verified that SRR diffusion can indeed increase the novelty and diversity of the generation process.

**Model Capacity Sensitivity Analysis.** We further study the effect of model capacity by increasing or decreasing the number of layers in the autoencoder and diffusion backbone (Table 4). Results show that changing the autoencoder depth only slightly alters quality, novelty, and diversity, indicating that encoding and decoding are relatively robust to capacity variations. In contrast, modifying the diffusion backbone has a more pronounced impact: adding a layer improves novelty and diversity, while removing a layer reduces both. This suggests that the diffusion model's capacity is more critical than that of the autoencoder, as it directly governs the ability to explore the latent space and balance quality with novelty.

Table 4: Ablation on model capacity. Param. Num. denotes parameter number.

| Approaches | Quality Scores | | | | Novelty Score | Diversity Score |
|---|---|---|---|---|---|---|
| | $S_{\text{sym}} \uparrow$ | $S_{\text{per}} \uparrow$ | $S_{\text{con}} \uparrow$ | Mean $\uparrow$ | $S_{\text{nov}} \uparrow$ | $S_{\text{div}} \uparrow$ |
| Increase AE Param. Num. | **0.753** | 0.471 | **0.977** | **0.734** | 0.308 | 0.411 |
| Decrease AE Param. Num. | 0.688 | 0.479 | 0.953 | 0.707 | 0.280 | 0.413 |
| Increase diff. Param. Num. | 0.705 | 0.474 | 0.936 | 0.705 | **0.330** | **0.444** |
| Decrease diff. Param. Num. | 0.722 | **0.493** | 0.955 | 0.723 | 0.286 | 0.409 |
| Original setting | 0.718 | 0.487 | 0.969 | 0.725 | 0.296 | 0.420 |

## 6 RELATED WORK

**3D Visual Content Generation.** Generative modeling of 3D structures has advanced rapidly with voxel-based autoencoders, implicit representations, and diffusion models. Early works such as voxel GANs and VAEs (Wu et al., 2016; Brock et al., 2016) produced coarse but plausible shapes, while point cloud and mesh models (Yang et al., 2019; Liu et al., 2023) improved geometric fidelity. Recent diffusion-based approaches (Nichol & Dhariwal, 2021; Guan et al., 2023; Mo et al., 2023) achieve state-of-the-art results in quality and diversity for generic 3D content. However, these methods primarily target visual plausibility, whereas metamaterials impose stricter constraints: generated designs must be mechanically valid and functionally novel. Thus, direct adoption of generic 3D generation is insufficient, motivating domain-specific frameworks that explicitly regulate and guide the design process.

**Metamaterial Generation.** AI-driven metamaterial discovery has explored multiple representations. Graph-based methods (Zhan et al., 2025; Xu et al., 2023; Zheng et al., 2023a) model unit cells as nodes and edges, which is well-suited for truss-based designs and property prediction but struggles with fine-grained geometry due to simplified primitives. 2D image–based approaches (Kollmann et al., 2020; Tian et al., 2022; Wilt et al., 2020) construct patterned planar motifs and extrude them along one axis, enabling strong in-plane performance but limited improvement along the extruded direction. Voxel-based approaches (Zheng et al., 2025; 2023b; Yang et al., 2024) offer a unified representation that can express different metamaterials like truss, shell or porous within a single discretization. Yet, current voxel generative models often face a quality–novelty trade-off: bias toward training data yields high-fidelity but unoriginal structures, while exploration leads to invalid designs.

## 7 CONCLUSION

In this paper, we introduced **VOXPLORER**, a framework for voxel-based metamaterial discovery that integrates latent space regulation with diffusion-based exploration. With RAS, we disentangle valid and invalid regions for robust decoding, and with SRR, we encourage exploration beyond memorized designs while preserving feasibility. We also introduced the first publicly available large-scale voxel dataset for truss-based metamaterials and a benchmark with metrics covering quality, novelty, and diversity. Experiments on MetaTruss and MetaShell show consistent gains over state-of-the-art baselines, confirming that VOXPLORER effectively balances quality, novelty, and diversity. This work lays the foundation for systematic study of next-generation metamaterials.

## REPRODUCIBILITY STATEMENT

We provide material to ensure that our work is fully reproducible. In particular, we provide the details on the model architecture, training scheme and benchmark in the Appendix. We will release our code and benchmark upon paper acceptance.

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

# A   MORE GENERATION RESULTS

## A.1   GENERATION RESULTS FOR METATRUSS

Figure 5 show some generated samples from all baselines and our model, which are trained on MetaTruss dataset.

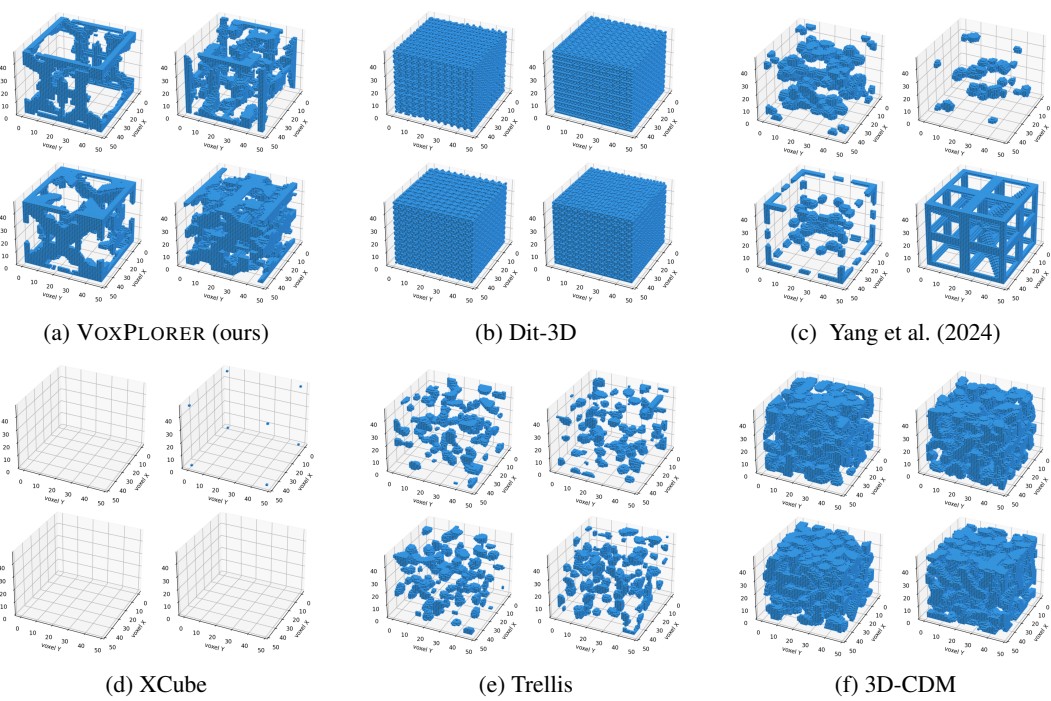

(a) VOXPLORER (ours)            (b) Dit-3D            (c) Yang et al. (2024)

(d) XCube            (e) Trellis            (f) 3D-CDM

Figure 5: Generated samples on MetaTruss with different models.

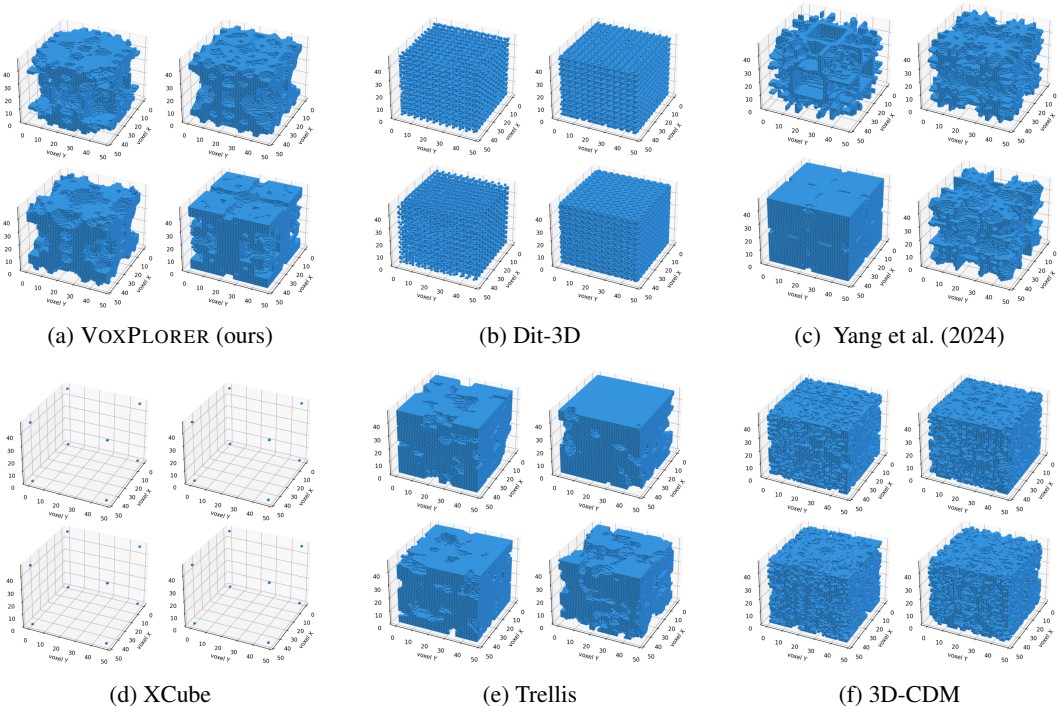

(a) VOXPLORER (ours)            (b) Dit-3D            (c) Yang et al. (2024)

(d) XCube            (e) Trellis            (f) 3D-CDM

Figure 6: Generated samples on MetaShell with different models.

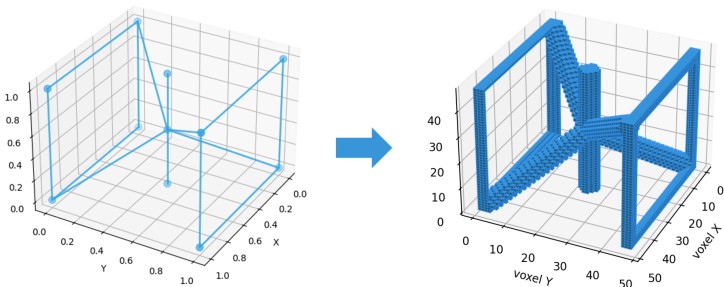

Figure 7: Data creation for MetaTruss.

### A.2 GENERATION RESULTS FOR METASHELL

Figure 6 show some generated samples from all baselines and our model, which are trained on MetaShell dataset.

**Remarks.** The generation results of our VOXPLORER have both improved novelty and genuine quality compared with other baselines. These results can serve as candidate novel designs for human experts to evaluate or find inspiration from.

## B DETAILS ON MODEL ARCHITECTURE AND TRAINING SCHEME

### B.1 MODEL ARCHITECTURE

The encoder and decoder we use are two transformers of the same structure, which have 4 layers and whose model dimension is by default 128. The latent space is also set to be 128-dimensional. The voxel data are decomposed into patches with the size of $8^3$, and flattened as the input to the encoder. After the encoder $\mathcal{E}$ and before the decoder $\mathcal{D}$, there is each an 2-layer multilayer-perceptron (MLP) to resize the data to and from 128-dimensional.

The latent diffusion model we use has a backbone of MLP which has 16 layers and a model dimension of 512, with residual links connecting adjacent layers.

### B.2 TRAIN SCHEME

The autoencoder is trained with RAS regulation. To enable this operation, we have to construct a negative dataset and combine it with the initial positive dataset. The negative data are created by noising each positive sample. We randomly select an eighth of the voxels and substitute them to void or Gaussian noise or an eighth of another positive sample, or simply add Gaussian to the initial values.

The diffusion model is trained following the DDPM paradigm, and the SRR mechanism only functions in inference stage.

## C DETAILS ON BENCHMARK

### C.1 DATASET CREATION AND REPRESENTATION UNIFICATION

Our dataset is created based on the dataset from Lumpe & Stankovic (2021), which comprises over 17,000 samples in 3D graph representation (left side of Figure 7). We select the first 10,000 samples from Lumpe & Stankovic (2021) and compute whether each voxel is close enough to any 3D edge in the 3D graph. If the distance is less than a predefined radius (e.g., 0.06), then the voxel is decided to be solid, or else the voxel is void. The distance $\delta$ between a voxel's center $\boldsymbol{c}$ an edge whose endpoints are $\boldsymbol{p}_1$ and $\boldsymbol{p}_2$ is:

$$\delta = \frac{||(\boldsymbol{p}_1 - \boldsymbol{c}) \times (\boldsymbol{p}_1 - \boldsymbol{p}_2)||}{||\boldsymbol{p}_1 - \boldsymbol{p}_2||}, \tag{11}$$

where $\times$ means outer product. Figure 7 gives an example of a structure before and after the above operation.

In the setting of our benchmark, the resolution of a voxel sample is determined to be 48. When constructing MetaTruss dataset, we directly make the shape of data to be $48^3$. The voxel data in MetaShell is initially of shape $128^3$. To unify the data representation, we use "skimage.transform.resize" to resize the voxel data into the needed dimension with interpolation.

## C.2 EVALUATION MODULE

The evaluation module of our benchmark systematically evaluates the voxel data from three aspects: quality, novelty and diversity. For quality, we are inspired from Chen et al. (2025) where the symmetry, periodicity and connectivity of the generated samples are calculated. However, the benchmark in Chen et al. (2025) is designed for graph-representation. In this paper we generalize the idea to voxel domain, and define the following three quality metrics:

$$S_{\text{sym}}(\mathbf{U}^{\text{gen}}) = 1 - \frac{\sum_{i,j,k=1}^{N} |u_{i,j,k}^{\text{gen}} - u_{N+1-i,N+1-j,N+1-k}^{\text{gen}}|}{2|\mathbf{U}^{\text{gen}}|}\}, \tag{12}$$

$$S_{\text{per}}(\mathbf{U}^{\text{gen}}) = \frac{1}{3}\left(\frac{\mathbf{U}|_{i=1}^{\text{gen}} \cap \mathbf{U}|_{i=N}^{\text{gen}}}{\mathbf{U}|_{i=1}^{\text{gen}} \cup \mathbf{U}|_{i=N}^{\text{gen}}} + \frac{\mathbf{U}|_{j=1}^{\text{gen}} \cap \mathbf{U}|_{j=N}^{\text{gen}}}{\mathbf{U}|_{j=1}^{\text{gen}} \cup \mathbf{U}|_{j=N}^{\text{gen}}} + \frac{\mathbf{U}|_{k=1}^{\text{gen}} \cap \mathbf{U}|_{k=N}^{\text{gen}}}{\mathbf{U}|_{k=1}^{\text{gen}} \cup \mathbf{U}|_{k=N}^{\text{gen}}}\right), \tag{13}$$

$$S_{\text{con}}(\mathbf{U}^{\text{gen}}) = \frac{\max_i |\mathbf{C}_i|}{|\mathbf{U}^{\text{gen}}|}, \tag{14}$$

where $S_{\text{sym}}$ measure the central symmetry degree, $S_{\text{per}}$ measures the periodicity degree, and $S_{\text{con}}$ measures the connectivity degree; $\mathbf{U}^{\text{gen}}$ is a generated sample in voxel representation, $\mathbf{C}_i$ is the $i$th cluster of connected voxels in $\mathbf{U}^{\text{gen}}$, $u_{i,j,k}^{\text{gen}}$ is a voxel in $\mathbf{U}^{\text{gen}}$ whose indices are $i, j, k$, $\mathbf{U}|_{i=1}^{\text{gen}}$ is a slice of voxels in $\mathbf{U}^{\text{gen}}$ whose the index along x axis is $i = 1$, and $|\mathbf{U}^{\text{gen}}|$ is the number of solid voxels in $\mathbf{U}^{\text{gen}}$.

For novelty we propose a distribution-based novelty score:

$$S_{\text{nov}}(\mathbf{U}^{\text{gen}}; \mathcal{U}_{\text{train}}) = 1 - \frac{\mathbf{U}^{\text{gen}} \cap \mathbf{U}_{\text{NN}}^{\text{train}}}{\mathbf{U}^{\text{gen}} \cup \mathbf{U}_{\text{NN}}^{\text{train}}}, \tag{15}$$

where $\mathbf{U}_{\text{NN}}^{\text{train}}$ is the nearest neighbor of $\mathbf{U}^{\text{gen}}$ in the training dataset $\mathcal{U}^{\text{train}}$.

For diversity we propose a distribution-based diversity score:

$$S_{\text{div}}(\mathcal{U}^{\text{gen}}; \mathcal{U}^{\text{train}}) = \frac{|\mathcal{L}|}{|\mathcal{U}^{\text{gen}}|}, \tag{16}$$

$$\mathcal{L} = \left\{l_i | l_i = \arg\max_{l' \in \{1,2,\cdots,|\mathcal{U}^{\text{train}}|\}} \frac{\mathbf{U}_i^{\text{gen}} \cap \mathbf{U}_{l'}^{\text{train}}}{\mathbf{U}_i^{\text{gen}} \cup \mathbf{U}_{l'}^{\text{train}}}, i \in \{1, 2, \cdots, |\mathcal{U}^{\text{gen}}|\}\right\}, \tag{17}$$

where $\mathbf{U}_i^{\text{gen}}$ is the $i$th elements in the set of generated samples $\mathcal{U}^{\text{gen}}$, $\mathbf{U}_{l'}^{\text{train}}$ is the $l'$th elements in $\mathcal{U}^{\text{train}}$.

## D MORE ABLATION RESULTS

In this section we provide some extra ablation results conducted on the MetaShell dataset.

Table 5: Ablation on RAS regulation and SRR diffusion.

| Approaches | Quality Scores | | | | Novelty Score | Diversity Score |
|---|---|---|---|---|---|---|
| | $S_{\text{sym}}$ ↑ | $S_{\text{per}}$ ↑ | $S_{\text{con}}$ ↑ | Mean ↑ | $S_{\text{nov}}$ ↑ | $S_{\text{div}}$ ↑ |
| Case 1 (RAS + vanilla DDPM) | **0.935** | **0.884** | 0.930 | 0.916 | 0.305 | 0.625 |
| Case 2 (w/o reg + SRR Diff.) | 0.910 | **0.795** | 0.842 | 0.849 | 0.237 | 0.477 |
| Case 3 (full framework) | 0.923 | 0.856 | **0.978** | **0.919** | **0.380** | **0.783** |

## E STATEMENT OF USE OF LARGE LANGUAGE MODELS (LLMS)

In this paper, we use LLMs only for polishing the words and checking for grammar errors. No technical details concerns the contribution of LLMs.

Table 6: Ablation on model capacity. Param. Num. denotes parameter number.

| Approaches | Quality Scores | | | | Novelty Score | Diversity Score |
|---|---|---|---|---|---|---|
| | $S_{\mathrm{sym}}\uparrow$ | $S_{\mathrm{per}}\uparrow$ | $S_{\mathrm{con}}\uparrow$ | Mean $\uparrow$ | $S_{\mathrm{nov}}\uparrow$ | $S_{\mathrm{div}}\uparrow$ |
| Increase AE Param. Num. | 0.915 | **0.858** | **0.985** | **0.919** | 0.362 | 0.711 |
| Decrease AE Param. Num. | 0.894 | 0.823 | 0.963 | 0.893 | 0.363 | 0.742 |
| Increase diff. Param. Num. | 0.920 | 0.810 | 0.953 | 0.894 | **0.389** | 0.781 |
| Decrease diff. Param. Num. | 0.907 | 0.852 | 0.956 | 0.905 | 0.359 | 0.775 |
| Original setting | **0.923** | 0.856 | 0.978 | **0.919** | 0.380 | **0.783** |

