# OpenReview forum: "Towards Novel Metamaterial Discovery via Latent Space Regulation and Exploration"
_ICLR.cc/2026/Conference — Submitted to ICLR 2026_

### Official Review · Reviewer_9WUB · 2025-10-30

**Soundness:** 2
**Presentation:** 1
**Contribution:** 1
**Rating:** 2
**Confidence:** 2

**Summary:**

This paper introduces both a benchmark for evaluating generative models generating metamaterials, and their own technique for performing this generation.

I think the paper is limited as it only considers methods based on the voxel-representation, and I also think there are plenty of things that can be improved regarding clarity in the paper. I also have some concerns regarding the numerical evaluation.

**Strengths:**

Ambitious with both new data and new model.

**Weaknesses:**

The paper focuses on voxel-based representations only. A benchmark should ideally not rely on the representations by the model, but be agnostic to such design choices.

I think the clarity can be improved (see Questions)

I think the paper is missing a discussion about the computational overhead with their model, and how the computation compares with other models.

**Questions:**

Why can’t you use graph-based methods, then convert their output to the voxel-based data (like you did when constructing the training dataset)?

Line 199: how do you measure distance to determine the “nearest neighbor” in the training set?

Line 239: how is the synthesization of negative samples performed? How large is this set of negative samples compared to the positive samples?

Eq 5+6: I guess you mean by pos/neg that this is either pos or negative, and then you have two terms (one for positive, one for negative)?

Regarding the evaluation, it is a bit concerning to me that multiple models in the MetaTruss benchmark have 0 novelty, and additionally, in those cases the quality scores are not similar. I do not understand exactly how the novelty score works, it seems to me that if this is 0, the generated samples should essentially overlap some sample in the training set. But still, for these models, the quality scores are very different. Any ideas why? Do you have some unseen real data to see what the metrics are for this data, and therefore what numbers one should expect from a good model?

Table 3: I get the feeling that the ablation is somewhat inconclusive as novelty is very poor with only SRR. I could understand it if the latent space has a lot of overlap, but shouldn’t SRR still be able to find something? To me, it seems like RAS is the bigger reason for improved novelty: without a good latent space, novelty will be poor. Also, could improve readability of the table if instead writing in the left column + RAS, + SRR Diff, + RAS + SRR Diff (full framework).

The ablation is missing the baseline no regularization, no SRR diffusion.

Line 462: you say the related methods provide a solution that is insufficient, but are there any empirical results that show that, or is this your own feeling/speculation? As far as I can see, you never try no regularization, no SRR diffusion. Or did someone else do this? In that case, a reference that backs up this statement is needed.

---

> ### Author Response · Authors · 2025-11-21
>
> We thank the reviewer for their time and constructive comments. Please find our detailed responses below.
>
> **Q1:** The paper focuses on voxel-based representations only?
>
> **A1:** We would like to clarify that voxel representation is a unified representation that can express arbitrary metamaterial structures. The problem we are studying is 3D metamaterial design in a unified representation, and that is why we build a corresponding benchmark.
>
> **Q2:** Why can’t you use graph-based methods, then convert their output to the voxel-based data?
>
> **A2:** Thank you for the question. Again, our goal is to develop a framework to cater the need of metamaterial design in a unified representation. However, graph-based models can only process graph data. Therefore, they cannot be applied to other data modalities such as voxel or shell-like metamaterials. This is why we don't compare model performance with graph-based models directly.
>
> **Q3:** How do you measure distance to determine the “nearest neighbor” in the training set?
>
> **A3:** As mentioned in line 199, we use intersection over union (IoU) to measure distance between two samples. The IoU calculates basically how much common voxels two structures share and how much different voxels they have between each other.
>
> **Q4:** Creation of negative samples.
>
> **A4:** Please refer to our response to Q2 of reviewer mpvG. Besides, we synthesize the same amount of negative data as the positive data, as mentioned with "by
> noising each positive sample" in the appendix (line 737).
>
> **Q5:** What does the "pos/neg" notation mean?
>
> **A5:** This notation means that equation 5 and 6 account for both the positive and negative samples. We are sorry that this notation causes some ambiguity and we would add some footnote into our paper to explain this.
>
> **Q6:** Concern on 0 novelty for some baseline models.
>
> **A6:** Like in our reply to Q7 if reviewer Ekph, we set a threshold to cut-off some invalid generated samples. Specifically, if one quality score of a structure is below a given threshold (0.5 in our setting), then this structure is considered invalid. Therefore, some results have 0 novelty. We will more clearly describe this operation in our revised paper.
>
> **Q7:** Novelty is very poor with only SRR.
>
> **A7:** Thank you for the insightful comment. The reason why this happens is related to the above Q6 and A6. Without RAS to disentangle the negative and positive latents, the area beyond the known positive latents is much more "invalid" than if the two types of latents are disentangled. Therefore, applying SRR without regulating the latent space can yield low-quality samples. Since we do not count the novelty of invalid low-quality samples, the novelty score for SRR alone becomes poor.
>
> **Q8:** The ablation is missing the baseline no regularization, no SRR diffusion.
>
> **A8:** We would like to argue that by comparing the three cases in Table 3, the effect of the two main modules of our framework can be studied. Still, we add another ablation study case where we use no regularization and the vanilla DDPM diffusion model. Result is shown below.
> |  | $S_\mathrm{sym}$ | $S_\mathrm{per}$ | $S_\mathrm{con}$ | Mean | $S_\mathrm{nov}$ | $S_\mathrm{div}$ |
> |---|---|---|---|---|---|---|
> |MetaTruss| 0.577 | 0.339 | 0.471 | 0.462 | 0.005 | 0.003 |
> |MetaShell| 0.660 | 0.698 | 0.925 | 0.761 | 0.188 | 0.104 |
>
> **Q9:** Empirical results to show that related methods are insufficient.
>
> **A9:** Please check the additional results in A8. The vanilla latent diffusion model (no regularization, no SRR mechanism) underperforms our framework (as compared with results in Table 1).
>
> We sincerely thank reviewer 9WUB again for the time and effort spent.

---

> > ### Comment · Reviewer_9WUB · 2025-11-25
> >
> > I thank the authors for their reply. However, I am still a bit concerned. First off, the authors say that the voxel-based representation is a **unifying** representation, but still some methods (graph-based) cannot process this. This is to me a contradiction, especially as the authors describe in C.1 how they used the graph-data and converted it to voxels. The point with “Q2” was that why can’t you train a model on the graph-data, generate graph-data, and then convert that graph to voxels like you did when creating the training data?
> >
> > I also think it is weird that the authors now mention that they have applied some (ad-hoc?) strategy to rule materials as invalid and not compute novelty on this. I don’t remember reading this in the paper, did I miss it?
> >
> > Regarding pos/neg, I read the appendix but still feels like it is lacking details, like, how does the Gaussian noise look like (mean (I guess 0) and covariance matrix)?
> >
> > Overall, I still think the paper lacks some clarity, I don’t understand why not using graph-based methods, and my comment on computational time went unnoticed. I keep my score.

---

### Official Review · Reviewer_MJqw · 2025-11-01

**Soundness:** 3
**Presentation:** 3
**Contribution:** 2
**Rating:** 4
**Confidence:** 3

**Summary:**

This paper presents VoxPlorer, which is a generative model for creating collections of diverse, high-quality metamaterial designs over a voxel representation. To overcome the hurdles of the voxel design space -- which contains large swaths of invalid designs -- this paper first reduces the dimensionality by using an autoencoder to project the voxel grid into a small latent space. This latent space is carefully regularized to encourage smooth sample distributions while enforcing separability between the valid and invalid subspaces. The curated latent space makes it possible for stable diffusion-based exploration of valid structures, even in regions that are not populated by known samples. This stability allows us to use a bolder sampling strategy guided by SRR, which explicitly pushes the sampling process away from known samples in latent space, into novel yet feasible design regions. The authors also offer a set of metrics for evaluating the current and future collection(s) of voxel-based metamaterial designs.

**Strengths:**

This paper addresses an interesting and challenging problem in metamaterial design -- namely, the generation and evaluation of large, valid metamaterial datasets. The regularization applied to the latent embedding space is insightful and grounded, while seemingly effective and widely generalizable. The ablation results are also compelling, particularly in the qualitative examples showing the impact of different regularization schemes (Table 2). The text is also very well written and illustrated.

**Weaknesses:**

1. The authors position the unified representation as a major motivating factor, but the experimental results reflect experiments over trusses and shells separately. Does VoxPlorer continue to function as designed in the presence of distinct classes? I'm particularly curious whether the latent space regularization and diffusion sampling can effectively perform their roles over a combined dataset.

2. Since the negative voxel samples are critical to your latent space construction, it would be nice to include more information about their definition and construction process in the main paper. The current supplemental section is also somewhat sparse. Did your noising process include any consideration for symmetry or periodicity? Did you perform any checks on the resulting "negative" samples to ensure that they are invalid as desired? What criteria are used to determine (in)validity? Relatedly, do you track the relative frequency of (in)valid designs generated by your models and/or other baselines?

3. The constraints of symmetry and periodicity are discussed several times (and feature prominently in your evaluation metric suite), but they do not seem to factor into the system design in any meaningful way -- not in the latent space, regularization, diffusion, or negative sample creation. I'm particularly curious about the relationship between symmetry/periodicity constraints and the latent space, as the former attributes offer two principled axes for dimensionality reduction, yet it seems that they are not being used. It also seems as if this choice is made in opposition to the approach of MetaShell [Yang et al. 2024], since they're explicitly enforcing cubic symmetry (and with it, periodicity). Is there a reason that you chose not to (or were unable to) leverage these structural constraints?

4. Methods like 3D-CDM are specifically designed for performance-guided inverse design, so I wonder how meaningful it is to compare their novelty/diversity metrics in an unconditional generation setting. Could the authors comment on this choice?

5. There are a few lines of related work that are not currently represented, but came to mind while I was reading. These specific citations are not required, but I share them here for the authors' consideration:
	- l. 54 -- A few recent works introduced other class-spanning metamaterial representations, including Xue et al. (2025, "MIND: Microstructure INverse Design with Generative Hybrid Neural Representation") and Makatura et al. (2023, "Procedural Metamaterials: A Unified Procedural Graph for Metamaterial Design")
	- l. 74 - There are large datasets/generative frameworks for shell-type metamaterials, including "Parametric Shell Lattices" [Liu et al. 2022] and "Data-Efficient Discovery of Hyperelastic TPMS Metamaterials with Extreme Energy Dissipation" [Perroni-Scharf et al. 2025].
	- l. 150 - there are quantitative metrics for evaluating metamaterial design diversity, such as those proposed by METASET [Chan et al. 2020] and others discussed in a recent survey by Lee et al (Data-Driven Design for Metamaterials and Multiscale Systems: A Review", 2023).
These papers do not invalidate the current submission, but they do temper some of the claims/contributions made in the paper.

**Questions:**

Please see weaknesses.

1. If symmetry is an integral part of metamaterial design, why should rotational augmentation as in l. 190 be relevant or useful?

## Minor comments
- l. 292 -- clapse --> collapse
- l. 344 -- exploartion --> exploration
- throughout (e.g., 4.2 header, Table 2 caption) -- "regulation" and "regularization" seem to be used interchangeably, but it should be exclusively the latter.
- Table 2 -- Consider reordering the rows so that rows 1-3 are in the same order as case 1-3 as discussed in the corresponding paragraph

---

> ### Author Response · Authors · 2025-11-21
>
> We thank the reviewer for the constructive feedback and insightful questions. Below are our detailed responses.
>
> **Q1:** The experimental results reflect experiments over trusses and shells separately.
>
> **A1:** We would like to clarify that a unified representation does not mean an integrated distribution of data. Despite that the representations of MetaTruss and MetaShell are the same, the distributions of data are different. Therefore, they are two datasets to benchmark various models. To make the experiment more comprehensive, we conduct further experiments to test our model on a combined dataset (of both MetaTruss and MetaShell), and the result is listed below.
>
> | $S_\mathrm{sym}$ | $S_\mathrm{per}$ | $S_\mathrm{con}$ | Mean | $S_\mathrm{nov}$ | $S_\mathrm{div}$ |
> |---|---|---|---|---|---|
> | 0.835 | 0.671 | 0.979 | 0.828 | 0.305 | 0.550 |
>
> According to the results above, our model performs reasonably in a combined dataset. Therefore, the representation can be regarded reasonably as a unified representation and our framework reasonably processes such a unified representation.
>
> **Q2:** Question regarding the creation of negative samples.
>
> **A2:** Please kindly refer to our response to Q2 of reviewer mpvG. We acknowledge that the current way of creating negative samples is to some degree still simple, but the experiments show the effectiveness of such a technique. We would like to thank reviewer MJqw for suggesting that we should add more details about this into the main body of our paper and we will do so in the later version of our paper.
>
> **Q3:** What does not VoxPlorer integrate symmetry and periodicity constraints into its framework via dimensionality reduction?
>
> **A3:** Our focus is to try to study how AI models can better capture physical principles in a data-driven way. Therefore, we do not specifically enforce for example symmetry via dimensionality reduction, but instead let the model learn it and test the result with the metric of symmetry score. However, such enforced constraints can be readily added into our framework just as done in Yang et al 2024. Overall, we are trying to develop AI models for novel metamaterial designs which should have certain physical constraints, but we don't want to be limited to specific physical rules, hence our current way of representing the data and building up the framework.
>
> **Q4:** How meaningful it is to compare their (methods like 3D-CDM) novelty/diversity metrics in an unconditional generation setting?
>
> **A4:** We appreciate this question. The task in our paper is to explore the geometrical space of voxel-represented metamaterial structures, instead of inverse design. We find two categories of potential baseline models, 3D structure generation models in computer vision domain and voxel-based metamaterial design models. Although models like 3D-CDM is primarily targeted for inverse design, they are still the closest baseline to our framework. Therefore, we think it is more proper to compare with them than not.
>
> **Q5:** The reviewer shares some related existing works.
>
> **A5:** Thank you very much for your valuable suggestion and information shared. We agree that a more comprehensive and detailed comparison with related works may help to make our paper more sound. However, just as reviewer MJqw mention, these extra existing works like existing datasets does not harm the fundamental contributions of our paper, though our discription and narrative may be better revised and adapted.
>
> **Q6:** If symmetry is an integral part of metamaterial design, why should rotational augmentation as in l. 190 be relevant or useful?
>
> **A6:** We would like to clarify that the rotational augmentation is not rotating the entire structure with an arbitrary angle. Instead, we rotate the cubic unit cell by 90 degrees in each rotation operation. This in essence shifts for example the facet of the x and y axes to x and z axes or so. Such augmentation can make the model to view the cubic unit cell from different axes, therefore the augmented structures can be "new" to the model while still be valid.
>
> Again, we thank reviewer MJqw for your valuable time and constructive feedback.

---

### Official Review · Reviewer_Ekph · 2025-11-04

**Soundness:** 3
**Presentation:** 3
**Contribution:** 2
**Rating:** 4
**Confidence:** 4

**Summary:**

The paper introduces, a generative framework aimed at advancing metamaterial discovery using voxel representations. Metamaterials are artificially engineered structures whose remarkable properties—such as negative Poisson’s ratio or high stiffness-to-weight ratio—stem from their geometry rather than composition. They are vital in applications ranging from soft robotics to energy absorption. The voxel representation, which discretizes 3D space into solid or void cells, is particularly advantageous for unifying diverse metamaterial classes like truss, shell, and porous designs. However, voxel-based generative models face the generalization dilemma, where the vast design space and limited datasets lead to models that either overfit to known designs (losing novelty) or generate invalid structures (losing quality).

To tackle this, integrates two key innovations: latent space regulation and guided exploration. First, it encodes complex voxel structures into a low-dimensional latent space using an autoencoder enhanced with a Repel-and-Sink (RAS) mechanism. RAS ensures a smooth, dense latent distribution while clearly separating valid and invalid regions, mitigating mode collapse. It consists of three parts: Inter-Class Repulsion (IeR), which simplifies the decision boundary between valid and invalid samples; Intra-Class Repulsion (IaR), which prevents latent clusters from collapsing; and a Central Sink (CS), which maintains distribution compactness by pulling latents toward the origin. Together, these elements enhance robustness and maintain structural validity.
In the generation phase, the work employs a diffusion model for sampling new designs, augmented by Short-Range Repulsion (SRR) guidance. SRR introduces a force that repels the current latent sample from known training latents, encouraging exploration of new but still feasible design regions. The repulsion decays rapidly with distance, ensuring exploration remains within valid boundaries defined by RAS. This combination allows VOXPLORER to balance the competing goals of novelty and validity.
To address the lack of benchmarks in voxel-based metamaterial design, the authors developed a comprehensive dataset and evaluation suite. They contributed MetaTruss, the first large-scale voxel dataset for truss-based metamaterials containing 60,000 samples discretized into a 48³ grid, alongside the existing MetaShell dataset for shell-type structures. Their evaluation framework employs five metrics across three dimensions: quality (symmetry, periodicity, and connectivity scores), novelty (IoU distance from nearest training sample), and diversity (variety of unique nearest neighbors among generated samples).

Experimental comparisons with state-of-the-art baselines like DiT-3D and 3D-CDM showed that VOXPLORER substantially outperforms existing models, achieving on average +8.9% improvement in quality, +46.4% in novelty, and +128.6% in diversity across datasets. On MetaShell, it matched or exceeded quality benchmarks while more than doubling diversity. Ablation studies confirmed that RAS effectively separates valid from invalid latents, while SRR significantly enhances novelty and diversity without degrading structure quality.
Overall, VOXPLORER presents a principled and scalable approach for discovering next-generation metamaterials. By uniting latent space regulation through RAS with guided exploration via SRR, it successfully achieves a robust balance between quality, novelty, and diversity in voxel-based generative metamaterial design.

**Strengths:**

The paper demonstrates remarkable strengths through its innovative generative framework, VOXPLORER, and its major contribution to the systematic benchmarking of voxel-based metamaterial design. Its first strength lies in offering a novel technical solution to the generalization dilemma (C1)—a persistent challenge in voxel-based generative models where methods often compromise between quality and novelty. VOXPLORER overcomes this trade-off with two ingenious mechanisms: the Repel-and-Sink (RAS) latent regulation, which introduces Inter-Class Repulsion, Intra-Class Repulsion, and a Central Sink to refine latent space organization and improve model robustness; and the Short-Range Repulsion (SRR) guidance, which ensures the exploration process avoids overfitting by pushing sampling away from memorized data while still generating valid, high-quality designs. These mechanisms collectively enhance generalization, producing novel yet feasible metamaterial structures.

**Weaknesses:**

Despite its impressive contributions, the paper also faces several weaknesses and practical challenges related to both the underlying voxel-based design field and the implementation of the VOXPLORER framework. A major limitation lies in the inherent challenges of voxel representation and data sparsity. The research domain itself suffers from an enormous design space—on the order of (2^{64^3}) possible configurations—while available training datasets remain extremely limited. Although the authors introduce the 60,000-sample MetaTruss dataset, voxel data remain costly to build and store, meaning that even the expanded dataset still represents a tiny and sparse portion of the possible design space. Consequently, VOXPLORER continues to operate under conditions of limited data coverage, which constrains the model’s ultimate generalization potential.

Another challenge arises from trade-offs in performance metrics. While VOXPLORER achieves a strong balance between quality, novelty, and diversity, these objectives are inherently competing. The Short-Range Repulsion (SRR) guidance successfully pushes the model toward unexplored regions of the latent space, boosting novelty and diversity; however, this exploration naturally comes with a modest deterioration in quality compared to standard diffusion-based methods. The paper acknowledges that this reduction is expected, as conventional Denoising Diffusion Probabilistic Models (DDPMs) tend to reproduce training samples—thus, by seeking novelty, VOXPLORER inevitably sacrifices a degree of structural refinement.

Overall, these weaknesses highlight both the technical challenges of scaling voxel-based metamaterial generation and the computational sensitivity of VOXPLORER’s design, underscoring that while the framework marks a substantial step forward, further refinement and domain adaptation remain essential for broader practical deployment.

**Questions:**

Thanks for the paper, it does seem like a good work to me, although I have some important questions which needs to be addressed. Kindly go through them.

1. [Section 1] : The authors should also note other method of material generation in general, they have only noted graph and voxel representation. I recommend the authors to include other representation space methods which involve grouping techniques. (Although maybe limited to crystals, but worth noting in this paper as related work). A few of these works can include 1. Xie, Tian, et al. "Crystal diffusion variational autoencoder for periodic material generation." arXiv preprint arXiv:2110.06197 (2021)., 2. Sinha, Anshuman, Shuyi Jia, and Victor Fung. "Representation-space diffusion models for generating periodic materials." arXiv preprint arXiv:2408.07213 (2024)., 3. Luo, Youzhi, Chengkai Liu, and Shuiwang Ji. "Towards symmetry-aware generation of periodic materials." Advances in Neural Information Processing Systems 36 (2023): 53308-53329.

2. [Line 71-92] : kindly include a bullet-wise details of the contribution of this work, simultaneously with each bullet refer the section where you have addressed your claim. This will make it more readable.

3. Kindly add motivation for RAS and SRR, has such a regularization technique never been seen in latent space models? If yes, then kindly also cite them properly.

4.  Generalization Dilemma (C1): What precisely is the Generalization Dilemma (C1) that VOXPLORER seeks to solve, and how does the combination of the vast design space (e.g., $2^{64^3}$ configurations) and limited dataset size (around 10,000 samples) cause this issue?

5. I have a couple of question regarding Table 1. Seems like the authors have not mentioned why 3D-CDM and various other methods have score poorly on Snov and Sdiv, while Voxplorer does so much better. But that's not seen on the Quality Score? Also what's the contrast between MethaTruss and MetaShell.  Why other methods are more competitive for MetaShell. While lag behind heavily in MetaTruss. Does it have anything to do with the inductive biases your current approach has? Which is likely the case, when you specifically introduced the repulsion function.

6. Regularization of latent space is definitely not a new topic of research, although all those regularization have come up with some limitation or the other. I request the authors to kindly add a good section on limitations, since it's quite evident with the results.

7. I strongly encourage the authors to provide a detailed study of hyper-maters in order to control the latent space regularization. I couldn't find explicit detail on this topic. Do the latent space regularization and sampling have any associated hyper-parameters with which they can produce controllable results on Novelty and diversity? If no, then I think this should be included (I did not had the time to go over the appendix, particularly because it is not in the main paper and as a reviewer I am only assigned the time to review the main section).

The only talk on hyper-parameters which I saw is on eq 8 and 9, however there is no ablation study on that, Is the ablation in table 4 addressing that? If yes, then it should've been properly mentioned.

8. I do understand that this paper does not necessarily need experimental validation to support the claim that there method does generate materials which are valid though first principle methods. Although I would subject that to other reviewers who are expert in ab-initio calculations to check whether this would require any experimental validation or not.

Thanks, kindly address the above questions. Happy to discuss further.

**Details Of Ethics Concerns:**

No ethics review required.

---

> ### Author Response · Authors · 2025-11-21
>
> We sincerely thank reviewer Ekph for the thoughtful comments and constructive suggestions. Please find our point-wise responses below.
>
> **Q1:** Limitation of voxel representation.
>
> **A1:** We understand reviewer Ekph's concern of the large design space and relatively limited amount of data. Despite the limitation of voxel representation as pointed out by the reviewer, we still regard voxel representation as a promising direction to study. The reason is that it is a straightforward type of representation that can unify all kinds of metamaterials, and it is easy to use, e.g., for property calculation. Although other representations like graph or mesh are theoretically also able to represent all kinds of structures, the transformation from other representations to graphs or meshes are not trivial (e.g., porous structures to graphs), compared with voxel representation. Therefore, we acknowledge the limitation of voxel representation, but we believe the bonus it brings makes it worth the cost.
>
> **Q2:** Trade-off between performance metrics.
>
> **A2:** We propose VoxPlorer as a heuristic solution to alleviate the competing effect, i.e., the trade-off between quality and novelty of metamaterial design. I understand that there can be more advanced tricks for finding novel structures, for example, varying the functions used to design shell-based/skeleton-based metamaterial. These, however, are usually designed for a specific type of metamaterials. Our current focus is a data-driven AI framework that tries to address the mentioned trade-off. Overall, we believe that for current AI methods to find novel design which is usually some long-tail cases in the design space, the novelty-quality trade-off still remains a key challenge, a challenge not specially for our VoxPlorer alone.
>
> **Q3:** "The authors should also note other method of material generation in general, they have only noted graph and voxel representation."
>
> **A3:** We sincerely appreciate this comment. Still, we would like to make some clarification. We discussed 2D image-based, graph-based representation and voxel-representation in section 1, because 2D image and graphs are, so far as we know, the two dominant representations used in AI for metamaterial design domain. The other existing works as suggested by reviewer Ekph (CDVAE, StructRepDiff, and SyMat) all in essence take the representation of graphs. The only difference is that in crystal designs, the "edges" in graphs are not so much cared about so they basically model the atom positions. In the view of metamaterial, they are all essentially graphs. The difference is that they use different AI techniques to process the crystal/periodic structures represented with atom positions/types etc. Still, even after the clarification we make above, we believe it can be beneficial to take a broader view of existing works. We will include the discussion above into our appendix.
>
> **Q4:** Bullet-wise details of the contribution of this work.
>
> **A4:** Thank you for this suggestion. We agree and believe this may make our paper more readable. Here's a bullet-wise list of our contributions:
> * We propose a novel generative framework for voxel-based metamaterial generation.
> * We develop a comprehensive and extensible benchmark for voxel-based metamaterial design.
> * We conduct extensive experiments which show the strong performance of our proposed VoxPlorer framework.
>
> We will readily add these contents into later version of our paper.
>
>
> **Q5:** "Kindly add motivation for RAS and SRR, has such a regularization technique never been seen in latent space models?"
>
> **A5:** Firstly, we believe we have already discussed the motivation of RAS in the first paragraph of section 4.2 and motivation of SRR in the first paragraph of section 4.3. Second, for the regularization techniques, we mention the VAE's regulation (basically Gaussian regularization) and contrastive learning's regularization in section 4.2, which we believe are most related existing techniques. However, we do agree that existing guidance methods for diffusion models (related to SRR) are also worth mentioning, like classifier-free guidance. We will add such guidance methods into our methodology discussion.

---

> ### Author Response · Authors · 2025-11-21
>
> **Q6:** What precisely is the Generalization Dilemma (C1), and how is it caused?
>
> **A6:** The "Generalization Dilemma" is essentially the trade-off between quality and novelty. In the aspect of phenomenon, the generative models tend to either repeat seen structures (Figure 5 in Appendix, the last result generated by Yang et al's model) or generate failed structures (Trellis's generation result in Figure 5). The second point in the reviewer's question is how vast design space together with limited data cause this dilemma. For generative model like VAEs or diffusion models, if we can model the design space smoothly, like in regression problems where we have abundant data points, then the models will generalize well and generate diverse and valid samples. However, as the reviewer also points out in Question Q1, for voxel data we only have limited amount of samples compared with all possible combinations of voxels. Therefore, the latent space cannot be easily modeled smoothly. That's why we propose VoxPlorer to make a first step towards modeling a smoother and denser latent space, via the RAS mechanism.
>
> **Q7:** Why does VoxPlorer ourperform other baselines with a large gap on Snov and Sdiv, but not so much on quality score?
>
> **A7:** We would like to clarify that higher quality does not guarantee higher novelty and diversity. For example, Figure 5(b) shows that Dit-3D generates many structures that are very similar. These structures are largely connected, symmetric and periodic, which lead to high quality score, but they are close to a very limited subset of the entire dataset, which leads to a low diversity score. Another factor to consider is that when calculating novelty score Snov, we require the three quality metrics (Ssym, Sper and Scon) to be above a given threshold (e.g., 0.5), to be considered a valid design. If a design is largely invalid, it is not proper to consider "novelty" of it. This consideration makes some models to have low novelty score. Overall, we would like to argue that higher quality does not necessarily lead to higher novelty and diversity, especially with only the higher mean value of quality metrics.
>
> **Q8:** Contrast between MetaTruss and MetaShell, and models' performance on these two datasets.
>
> **A8:** Similar to other domains like computer vision, in the domain of voxel-based metamaterial design, there are also "easy" dataset and "hard" dataset. We consider MetaTruss to be a much harder dataset to model than MetaShell. The reason is that MetaTruss originates from graph representation, which has a very diverse distribution (please refer to "Thomas S Lumpe and Tino Stankovic. Exploring the property space of periodic cellular structures based on crystal networks. Proceedings of the National Academy of Sciences, 118(7):
> e2003504118, 2021." for more information on this). On the other hand, MetaShell is originates from varying the trigonometric functions used to define the shell-like structures, which is much less diverse than the truss structures, meaning they are similar in nature. The difference between these two datasets can also be partially observed from Figure 5 and 6. The distribution of voxels are less diverse for MetaShell than MetaTruss. To sum up, other baseline models perform better on MetaShell than they do on MetaTruss not because MetaTruss is biased towards the strength of VoxPlorer, but because MetaTruss is a harder dataset than MetaShell. We will readily add more details and comparison of the two datasets into the appendix to make our paper clearer.
>
> **Q9:** Add a good section on limitations of existing works that regularize of latent space.
>
> **A9:** We would like to thank the reviewer to give us this valuable suggestion. For the current version we only talk about Gaussian regularization in VAE and contrastive regularization. We will add more details into the appendix, to also compare other latent regularization works like Barbato, Francesco, et al. "Latent space regularization for unsupervised domain adaptation in semantic segmentation.", Hadjeres, Gaëtan, Frank Nielsen, and François Pachet. "GLSR-VAE: Geodesic latent space regularization for variational autoencoder architectures.", etc.
>
> **Q10:** Provide a detailed study of hyperparameters in order to control the latent space regularization.
>
> **A10:** Please kindly check our reply to Q3 of reviewer mpvG. Thanks for your suggestion.
>
> We would like to thank reviewer Ekph again for the very detailed comments and constructive suggestions towards making our paper better, and we would be glad to discuss further if there's anything we have still not made clear to the reviewer.

---

> > ### Comment · Reviewer_Ekph · 2025-11-25
> > **Kindly update the bullet enumeration to match the questions.**
> >
> > My final questions are clearly mentioned in the section question, they are numbered 1 to 8. You should answer in that order.

---

> > > ### Comment · Reviewer_Ekph · 2025-11-25
> > > **Rebuttal responses**
> > >
> > > Thank you authors for your responses. The numbers against my comment are for my reference while determining final evaluation of this paper and also to let the author know which comment are positive, which are negative and which are neutral. Kindly not worry about these. Thanks
> > >
> > > Response to A3.
> > > How are you defining the similarity between graph-based methods and representation based methods? As far as I know and you can read on this, representations are usually constructed with some material dependent physics with which you're able to get better representation. They are often addressed as being more useful that voxel based methods, with the focus on 'curse of dimensionality' in open generation of materials. I suggest you can read such works. I don't find your clarifications satisfactory. (-)
> > >
> > > Response to A4.
> > > Are you able to update the submission? If yes, then why did you not update it right away? If no, then I can ask the chairs for this. The usual way for all the previous ICLR years have been, you update the work and then properly cite the lines where you've made those changes. (-)
> > >
> > > Response to A5.
> > > Same as response to A4. If you have to add something then add it and then respond (-)
> > >
> > > Response to A7.
> > > May you kindly give us a brief overview on how are you calculating novelty and diversity? What is the denominator which you're dividing? And why do you say that it has a threshold on something? I don't see any score (in general) getting better or worse with such thresholds. Is your threshold not related to your score (in that case only that can happen). Will removing that threshold make the other models close the gap between you and them? (+)
> > >
> > > Response to A9
> > > Same as response to A4. If you have to add something then add it and then respond (0)
> > >
> > > Looking forward to your responses. Thanks again for the work.

---

### Official Review · Reviewer_mpvG · 2025-11-04

**Soundness:** 2
**Presentation:** 2
**Contribution:** 2
**Rating:** 4
**Confidence:** 3

**Summary:**

This paper addresses the challenge of generating novel yet valid 3D metamaterial designs in voxel space, where existing generative models either overfit or fail to maintain validity. The authors propose VOXPLORER, a two-part framework consisting of Repel-And-Sink (RAS) latent regulation to disentangle valid and invalid latent regions and Short-Range Repulsion (SRR) guidance during latent diffusion to promote exploration without sacrificing quality. They also introduce MetaTruss, a new voxel-based dataset, and unified metrics evaluating quality, novelty, and diversity. Experiments on MetaTruss and MetaShell datasets demonstrate that VOXPLORER achieves superior performance, improving quality by 8.9%, novelty by 46.4%, and diversity by 128.6% on average compared to state-of-the-art baselines.

**Strengths:**

1. The proposed method demonstrates superior performance compared to baseline approaches across two benchmark datasets.

2. The authors introduce a new voxel dataset for trusses (MetaTruss).

**Weaknesses:**

1. Lack of physics-based validation. The quality / validity evaluation relies only on voxel heuristics but not effective property evaluation (e.g., energy absorption, poisson's ratio).

2. Insufficient details on negative sample synthesis. The authors do not clearly explain how the negative samples are synthesiszed. The negative samples might not reflect realistic invalid metamaterial failure modes.

3. Incomplete hyperparameter analysis. The proposed method contains numerous hyperparameters. The authors do not describe how the hyperparameters are selected or provide a study to evaluate the sensitivity of the method's performance to the hyperparameters.

**Questions:**

See Weaknesses

---

> ### Author Response · Authors · 2025-11-21
>
> We sincerely thank reviewer mpvG for their thoughtful feedback. Please find our responses below:
>
> **Q1:** Lack of physics-based validation.
>
> **A1:** We appreciate the reviewer’s concern. The main focus of our paper is on the geometry of metamaterial structure, rather than inverse design, which is another important task, though. Therefore, our validation metrics also take the focus of geometrical quality and diversity.
>
> **Q2:** Insufficient details on negative sample synthesis.
>
> **A2:** We would like to clarify that some details related to the creation of negative samples are provided in Appendix B.2, though we acknowledge that more detailed description may be helpful. We also understand the concern from reviewer mpvG that synthesized samples may not be able to thoroughly reflect the real failure cases. More advanced synthesis tricks can be the focus for future works. Still, our ablation study shows that with the negative samples synthesized in our current way, the model performance is improved by a large margin (Case 2 vs. Case 3 in Table 3).
>
> **Q3:** Incomplete hyperparameter analysis.
>
> **A3:** Thank you for the suggestion. There are two main types of hyperparameters in our method. The first type is model capacity-related ones such as number of layers and latent dimension. We conduct experiment of the model capacity by varying the number of layers in the autoencoder and diffusion model. The results do not show an obvious tendency towards superiority of more or less model capacity. Another type is loss-related hyperparameters, or the three "lambda" factors in equation 9 and 10. We set $\lambda_\mathrm{recon}$ to be 1, and $\lambda_\mathrm{RAS}$ to be 0.001, and $\lambda_\mathrm{SRR}$ to be 0.01. Again, the model performance does not show an obvious favor towards larger or smaller $\lambda_\mathrm{recon}$ or $\lambda_\mathrm{RAS}$ so long as the order stays similar to the original setting.
>
> Again, we thank the reviewer mpvG for your time and effort in the review process.

---

### Meta-Review · Area_Chair_8yFA · 2026-01-10

**Summary:**

The main concerns are about methodological clarity and positioning:
- the choice of voxel representations and justification benchmarking against graph/representation-space methods,
- insufficient clarity and transparency in evaluation protocols,
- negative sample construction and latent regularization hyperparameters remaining under-specified in the main paper,
- missing discussion of computational cost, and
- limited engagement with related work and limitations within the main text.

**Reviewer Concerns:**

Concerns addressed:
- Clarification of methodology and design choices.
- Improved explanation of evaluation metrics and protocols.
- Negative samples and latent-space formulation clarified.
- Hyperparameters and ablation studies elaborated.
- Dataset behavior and generalization reasonably demonstrated.

Concerns remain:
- Evaluation rigor and metric clarity remain insufficient.
- Negative sample design and realism remain unresolved.
- Methodological scope and representation choices are not fully justified. Reviewers are unconvinced by the “unified representation” claim.
- Limited analysis of model behavior and design trade-offs.
- Manuscript clarity and rebuttal execution issues still persist

**Reviewer Scores:**

Reviewer 9WUB objected acceptance and did not increase their score. Other reviewers are likely to remain unchanged.

---

### Decision · Program_Chairs · 2026-01-26

Reject